# DNA Methylation Levels of the *TBX5* Gene Promoter Are Associated with Congenital Septal Defects in Mexican Paediatric Patients

**DOI:** 10.3390/biology11010096

**Published:** 2022-01-08

**Authors:** Esbeidy García-Flores, José Manuel Rodríguez-Pérez, Verónica Marusa Borgonio-Cuadra, Gilberto Vargas-Alarcón, Juan Calderón-Colmenero, Juan Pablo Sandoval, José Antonio García-Montes, Víctor Manuel Espinoza-Gutiérrez, Juan Gerardo Reyes-García, Benny Giovanni Cazarín-Santos, Antonio Miranda-Duarte, Armando Gamboa-Domínguez, Nonanzit Pérez-Hernández

**Affiliations:** 1Departamento de Biología Molecular, Instituto Nacional de Cardiología Ignacio Chávez, Ciudad de Mexico 14080, Mexico; egarciaf1905@alumno.ipn.mx (E.G.-F.); jose.rodriguez@cardiologia.org.mx (J.M.R.-P.); gilberto.vargas@cardiologia.org.mx (G.V.-A.); bcazarins1800@alumno.ipn.mx (B.G.C.-S.); 2Sección de Estudios de Posgrado e Investigación, Escuela Superior de Medicina, Instituto Politécnico Nacional, Ciudad de Mexico 11340, Mexico; jgreyesg@ipn.mx; 3Departamento de Medicina Genómica, Instituto Nacional de Rehabilitación Luis Guillermo Ibarra Ibarra, Ciudad de Mexico 14389, Mexico; vborgonio@inr.gob.mx (V.M.B.-C.); amiranda@inr.gob.mx (A.M.-D.); 4Departamento de Cardiología Pediátrica, Instituto Nacional de Cardiología Ignacio Chávez, Ciudad de Mexico 14080, Mexico; juan.calderon@cardiologia.org.mx; 5Laboratorio de Hemodinámica e Intervención en Cardiopatías Congénitas, Instituto Nacional de Cardiología Ignacio Chávez, Ciudad de Mexico 14080, Mexico; juanpablo.sandoval@cardiologia.org.mx (J.P.S.); antonio.garcia@cardiologia.org.mx (J.A.G.-M.); 6Hospital General de Zona No. 18, Playa del Carmen 77730, Mexico; dr.espinozacp@gmail.com; 7Departamento de Patología, Instituto Nacional de Ciencias Médicas y de la Nutrición Salvador Zubirán, Ciudad de Mexico 14000, Mexico; armando.gamboad@incmnsz.mx

**Keywords:** congenital septal defects, DNA methylation, *TBX5* gene, epigenetic markers, transcriptional factors

## Abstract

**Simple Summary:**

One of the most important health problems in the paediatric population, due to its prevalence worldwide, is the occurrence of congenital heart defects. The integration of multidisciplinary approaches to clinical diagnosis will allow us to detect this type of problem in patients in a timely way. The *TBX5* gene has an important participation in cardiogenesis. Therefore, we performed a case-control study, that involved the DNA methylation assessment of the *TBX5* gene promoter region in patients that were non-syndromic with congenital septal defects, to identify an epigenetic marker. Moreover, we evaluated the exposure to environmental factors during pregnancy in mothers of these patients. Additionally, we used bioinformatic tools to identify transcription factors binding to the *TBX5* gene region of interest and to learn their possible functional effect. These results could help to clarify the mechanisms that regulate these pathologies and establish risk markers, which can be used in future in clinical practice.

**Abstract:**

The *TBX5* gene regulates morphological changes during heart development, and it has been associated with epigenetic abnormalities observed in congenital heart defects (CHD). The aim of this research was to evaluate the association between DNA methylation levels of the *TBX5* gene promoter and congenital septal defects. DNA methylation levels of six CpG sites in the *TBX5* gene promoter were evaluated using pyrosequencing analysis in 35 patients with congenital septal defects and 48 controls. Average methylation levels were higher in individuals with congenital septal defects than in the controls (*p* < 0.004). In five CpG sites, we also found higher methylation levels in patients than in the controls (*p* < 0.05). High methylation levels were associated with congenital septal defects (OR = 3.91; 95% CI = 1.02–14.8; *p* = 0.045). The analysis of Receiver Operating Characteristic (ROC) showed that the methylation levels of the *TBX5* gene could be used as a risk marker for congenital septal defects (AUC = 0.68, 95% CI = 0.56–0.80; *p* = 0.004). Finally, an analysis of environmental factors indicated that maternal infections increased the risk (OR = 2.90; 95% CI = 1.01–8.33; *p* = 0.048) of congenital septal defects. Our data suggest that a high DNA methylation of the *TBX5* gene could be associated with congenital septal defects.

## 1. Introduction

Congenital heart defects (CHDs) are a group of complex diseases defined as developmental defects in the human heart; they represent an important problem of public health worldwide, with a prevalence of 4–50 per 1000 live births [1]. The development of CHDs are the result of an interaction between genetic and environmental factors. In the last decade, several studies have focused on genes involved in the development of heart defects, such as the *T-box* gene family and have found clinical evidence that this gene family contributes to the development of CHDs [2,3,4,5].

The *T-box* genes encode a family of transcription factors that share similar sequences within the DNA-binding domain (T-domain). In vertebrates, five subfamilies of genes involved in cardiogenesis have been identified: T family, Tbx1, Tbx2, Tbx6, and Tbr1 [6,7]. Recently, genes that belong to the Tbx1 (*TBX1*, *TBX18*, and *TBX20*) and Tbx2 (*TBX2*, *TBX5*) subfamilies have been reported as transcriptional activators and gene repressors involved in the formation of chamber myocardium. These genes have also been associated with the development of CHDs, including atrial septal defect (ASD), ventricular septal defect (VSD), mitral valve disease (MVD) and tetralogy of Fallot (TOF), among others [5,8,9,10,11].

In particular, the *TBX5* gene is a preponderant transcriptional activator of genes related and required for morphological changes in the septation and in cardiac conduction system [11]. The *TBX5* gene interacts with other genes in signalling pathways that modulate atrial septation; this finding is supported with functional evidence that shows its participation in the ontogeny of CHDs [12]. In addition, it has been reported that the *TBX5* gene interacts with the nucleosome remodelling and deacetylase (NuRD) repressor complex. This TBX5-NuRD complex is necessary for the normal cardiac development, therefore, any alteration could lead to CHDs [13].

On the other hand, it has been demonstrated that the *TBX5* gene is modified by different epigenetic mechanisms, including long non-coding RNAs (lncRNAs), histone modification and polycomb group proteins. These epigenetic modifications play a key role in the pathophysiology of CHDs [14,15,16,17]. Furthermore, some studies have shown that an imbalance in the status of DNA methylation could change the expression of genes that participate in cardiogenesis [8,10,18,19,20].

Nevertheless, the epigenetic mechanisms of the *TBX5* gene associated with congenital septal defects have been scarcely studied. Thus, we investigated the association of DNA methylation levels of the *TBX5* gene promoter with the risk of developing congenital septal defects. We also, analysed the association of methylation levels of *TBX5* gene with the exposure to environmental factors in mothers of patients with septal defects. Finally, we performed an in silico analysis to predict the possible functional role of the CpG sites studied by pyrosequencing.

## 2. Materials and Methods

### 2.1. Individuals Studied

This study included a total of 83 participants (35 patients with congenital septal defects and 48 controls). The selection criteria for this study were individuals of both sexes, including newborn to eighteen years, with a single diagnosis non-syndromic of ASD, VSD and patent ductus arteriosus (PDA); the exclusion criteria were patients with another type of CHD, other types of heart malformations and other types of infectious heart diseases, additionally, all participants were assessed by expert clinical geneticists with the aim to exclude syndromic forms of CHD.

For this study, we integrated these two types of CHDs (ASD and VSD) as a group of patients, even though both congenital septal defects seem to be different diseases, both conditions occur in related territories, they share the same characteristic of an incomplete septal closure and the same potential aetiology. Therefore, both malformations lead to similar clinical complications, for example, pulmonary hypertension, cerebral infarction, and arrhythmias [21,22]. These individuals came from different regions of Mexico and were attended in the Paediatric Cardiology Department at the “Instituto Nacional de Cardiología Ignacio Chávez” in Mexico City. The study complied with the Declaration of Helsinki and was approved by institutional ethical and research committees (Register No. 19-1138).

ASD was defined as a failure to close the communication between the right and left atria. VSD was defined as a failure in hemodynamic mechanisms and disturbance in the communication between left and right ventricles. In both cases, the congenital defects can lead to pulmonary hypertension, stroke and dysrhythmias [21,22]. The control group included individuals with a diagnosis of PDA, which was defined as a non-septal structure formed during foetal development, located between the pulmonary artery and the descending aorta, and usually undergoes spontaneous closure after birth [23]. 

The diagnosis of septal defects in all patients was made using echocardiography characteristics; additionally, clinical and physical examinations, as well as chest X-rays were taken into consideration. Information regarding environmental risk factors that could predispose to the development of septal defects (drug addiction, diseases during pregnancy, exposure to pollutants, infections during pregnancy, medications and vitamins intake) were obtained from the mothers of these patients during gestation. These data are presented in the Appendix A.

### 2.2. DNA Extraction and Bisulfite Treatment

DNA was isolated from 3–5 mL of peripheral blood samples using the Gentra Pure DNA isolation kit (QIAGEN, Hilden, Germany); the concentration and purity of the DNA extracted were calculated using a nanodrop 2000 spectrophotometer (Thermo Fhiser Scientific, Waltham, MA, USA). Genomic DNA was stored at −20 °C prior to bisulfite modification. A total of 500 ng DNA went into bisulfite modification using the Epitect Bisulfite Conversion Kit (QIAGEN, Hilden, Germany) following the manufacturer’s instructions, each sample of bisulfite-modified DNA was suspended in 20 μL of TE buffer for the following pyrosequencing assay.

### 2.3. DNA Methylation

The CpG island region (chr12: 114409768-114409805) in the genomic sequence GRGh38.p10 of the *TBX5* promoter was selected for PyroMark CpG assays to quantify site-specific methylation levels. This region included 6 CpG sites, represented in bold in the following sequence (**CGCG**CTGTAGC**CG**GCTC**CG**GAGTTTACTGCC**CG**AA**CG**A) from base −3623 to base −3661 relative to the transcription start site (TSS) (Figure 1).

PCR was performed using PyroMark PCR Kits (QIAGEN, Hilden, Germany) following the standard manufacturer’s protocol. A total volume of 25 μL was amplified with 25 ng bisulfite-converted DNA, 0.2 μM forward and reverse primers were incorporated, 12.5 μL of PyroMark PCR master mix 2X and 2.5 μL of Coral load concentrate 10X.

Thermal cycling conditions included an initial denaturation at 95 °C for 15 min, followed by 45 cycles of 94 °C for 30 s, 56 °C for 30 s and 72 °C for 30 s, followed by a final extension of 72 °C for 10 min to obtain an amplicon length of 227 bp. The product quality of PCR was confirmed using 1.5% agarose gels with ethidium bromide staining.

Then, by the processes of PCR and pyrosequencing, the sequence, after bisulfite treatment (YGYGTTGTAGTYGGTTTYGGAGTTTATTGTTYGAAYGA), was analysed with the Hs_TBX5_04_PM 148 PyroMark CpG assay (GeneGlobe ID: PM000509609) commercially available from the supplier (QIAGEN, Hilden, Germany). Pyrosequencing was performed to detect the methylation levels of CpG sites through PyroMark Q24 (QIAGEN, Hilden, Germany), wherein PCR products were used to prepare single-stranded DNA templates combined with the pyrosequencing primer following the manufacturer’s protocol. Each pyrosequencing experiment included an EpiTect DNA methylated and unmethylated control (QIAGEN Cat. Num. 59695), as well as a negative control. The pyrosequencing quality control was assessed for each sample using PyroMark Q24 Advanced Software V.3.0.0, which allowed us to analyse and visualize the methylation levels expressed as a percentage.

### 2.4. In Silico Analysis

The prediction of possible functional effects for the *TBX5* promoter region, was analysed with PROMO, TFBIND and AliBaba 2.1 software [24,25,26] which identify and predict transcription factors binding affinities to DNA region evaluated by pyrosequencing.

### 2.5. Statistical Analysis

For quantitative variables, data were expressed as medians and interquartile ranges, whereas qualitative variables were expressed as frequencies and percentages. The comparison between categorical variables was obtained by a χ^2^ test. To evaluate the methylation levels of each CpG site between the study groups, the Mann–Whitney U test was applied. To determine the correlations between methylation levels of the *TBX5* gene and age, the Spearman test was performed. The association of environmental factors with methylation levels was calculated by univariate logistic regression. Methylation levels of the CpG sites were classified as quartiles; the lowest quartile was used as a reference group with the aim to calculate the risk of septal defects according to the distribution of methylation levels by a model of logistic regression. A Receiver Operating Characteristic (ROC) curve was plotted, the area under the curve (AUC) and their confidence intervals were estimated for the prediction capacity. The statistical analysis was two-tailed, and the alpha level of significance was <0.05. The analysis was performed using the SPSS 22.0 Software (SPSS Inc., Chicago, IL, USA) and GraphPad Prism Software 6.01 (GraphPad Software, La Jolla, CA, USA).

## 3. Results

We included 35 non-syndromic paediatric patients with septal congenital defects, 48.6% were females and 51.4% were males with a median age of 8 years [IQR 4–13]. For the control group, we included 48 individuals of whom 70.8% were females and 29.2% were males, with a median age of 3 years [IQR 2–6]. The population study characteristics are described in Table 1.

### 3.1. DNA Methylation of the TBX5 Gene Promoter and Its Association with Septal Defects

Six CpG sites in the promoter region of the *TBX5* gene were analysed. We included an example of a pyrogram result, which contains the methylation quantification of each CpG site expressed as a percentage, in Figure 2.

We found significant differences in the levels of methylation represented as median [interquartile range] between the patient and control groups. The average methylation level of all CpG sites was significantly higher in patients with congenital septal defects 7.6% [IQR 7–8.5%] than in the controls 6.3% [IQR 6–7.6%], *p* < 0.004. The methylation level of five CpG sites was significantly higher in patients with congenital septal defects than in the controls, with a median value at site 1: 7% [IQR 6–8%] vs. 6% [IQR 6–7%] *p* = 0.032; site 2: 6% [IQR 5–7%] vs. 5% [4–6%] *p* = 0.001; site 3: 7% [IQR 7–8%] vs. 6% [IQR 5–7.7%] *p* = 0.004; site 4: 10% [IQR 9–12%] vs. 8% [IQR 7–10.7%] *p* = 0.004; site 5: 11% [IQR 10–14%] *p* = 0.001. In site 6, no differences were observed (Figure 3).

Additionally, we analysed whether the methylation levels of all CpG sites confers a risk in both study groups, by stratifying the methylation levels into quartiles. We observed a significant increased risk of developing septal defects in the highest quartile [OR (95% CI) = 3.91 (1.02–14.8); *p* = 0.045) (Table 2).

Then, we performed a ROC curve analysis to the discriminatory value of the methylation levels of the *TBX5* gene in patients with septal defects when comparing to the control group; the result showed an AUC of 0.685 (95% CI = 0.568–0.803; *p* = 0.004) (Figure 4).

We also evaluated the association between environmental factors of mothers of patients with CHD during pregnancy and methylation levels of the *TBX5* gene. We observed that maternal infections during pregnancy were associated with an increased risk of development septal defects [OR (95% CI) = 2.90 (1.01–8.33); *p* = 0.048] (Table 3).

Correlations between methylation levels and age or sex were then analysed. No correlation with age was found, either in controls (r = 0.14, *p* = 0.33), or in patients with congenital septal defect (r = 0.16, *p* = 0.35). Likewise, no correlation between DNA methylation levels and sex were observed: control group females (r = 0.28, *p* = 0.10), males (r = 0.28, *p* = 0.39); patient group females (r = 0.17, *p* = 0.45), males (r = 0.38, *p* = 0.11). (Appendix A).

### 3.2. In Silico Analysis of Transcription Factor Binding Sites to TBX5 Promoter Region

The in silico analysis with PROMO, TFBIND and AliBaba 2.1 (http://gene-regulation.com/pub/programs/alibaba2/index.html accessed on 26 September 2021) software for the possible prediction of potential transcription factors binding sites to the *TBX5* promoter region indicated the binding of Sp1 (Specificity protein 1) and p53 (Tumor suppressor Trp53) transcription factors. See Figure 5.

## 4. Discussion

To date, the epigenetic involvement of the *TBX5* gene in CHDs is poorly understood. Thus, in this research, a quantitative DNA methylation analysis of the *TBX5* gene promoter in non-syndromic patients with congenital septal defects was performed. Additionally, the mothers of individuals with septal defects were analysed with regard to their exposure to environmental risk factors during pregnancy; likewise, we made a bioinformatics analysis in the promoter region of the *TBX5* gene.

Our results showed significant differences in methylation levels when comparing patients with congenital septal defects to the controls, which suggests that the *TBX5* gene could be a marker for risk association with septal defects. This assertion is based on the results of the ROC analysis, suggesting that methylation levels may be an epigenetic marker to detect individuals with congenital septal defects.

As far as we know, this the first study with an approach that included congenital septal defect patients and the *TBX5* gene; however, various investigations have been carried out in patients with TOF, with different genes implicated in cardiogenesis. TOF is a frequent type of CHD with implications in four different malformations such as overriding aorta, pulmonic stenosis, ventricular septal defect (VSD) and right ventricular hypertrophy. Therefore, we consider it pertinent that TOF and the pathology included in our study share relationships such as septal defects, although some authors include other relevant genes during the cardiogenesis [20,27,28,29,30].

For instance, Sheng et al. analysed the methylation levels of 71 candidate genes for CHDs in 41 patients with TOF and six controls. The authors found significant differences in 26 genes and within these, the *TBX5* gene showed increased methylation levels in patients with TOF when compared to the controls [27]. Additionally, Zhang et al. reported methylation levels of the *RXRA* gene promoter in 26 patients with TOF and in six controls by means of quantitative methods. The authors found that the overall methylation levels in the promoter region, was significantly higher in patients with TOF than in the controls [28]. In another research, Yuan et al. studied the methylation status of the *VANGL2* gene in 15 patients with TOF and in 5 controls. Their results showed that overall methylation levels of the *VANGL2* gene were significantly higher in patients with TOF than in the controls [20]. These results are in agreement with our findings, as we found that patients also presented higher methylation levels than the controls.

Evidence has shown that the foetus development could be influenced alterations in maternal DNA methylation; in this sense, the uterine microenvironment has the capacity to induce important changes in the global DNA methylation and can result in direct changes in gene expression in the developing foetus [31,32,33,34,35]. Therefore, we assessed environmental and lifestyle exposures factors during pregnancy in mothers of individuals with CHDs to understand this relationship. We observed differences in DNA methylation levels in individuals with CHDs when the mothers were exposed to harmful environmental factors during pregnancy. This evidence suggests at least in part, the participation of these epigenetic events in CHDs; nevertheless, the molecular mechanisms of these phenomena are still not fully understood.

The results of in silico analysis showed that modifications in the promoter region of the *TBX5* gene produced binding sites for the transcription factors Sp1 (Specificity protein 1) and p53 (Tumor suppressor Trp53). Factor p53 is implicated in the transrepression and transactivation of multiple genes. The increased level of p53 expression contributes mainly to cell cycle arrest, apoptosis, metabolism disorder and autophagy; it also plays a role in some cardiovascular diseases and CHDs, as it is required for embryonic heart development. During the embryonic and post-natal development, an inappropriate p53 activation can disrupt the development of the heart by initiating apoptosis and limiting cell proliferation [36,37]. Moreover, under physiological conditions, p53 participates in providing support to the cardiac structure, as well as in functions related to mechanisms of DNA repair, cell division, and cellular senescence [38,39].

Sp1 is a transcription factor implicated in several biological processes including cell growth, cell differentiation and chromatin remodelling [40]. This factor can increase or decrease the transcriptional activity, because of its affinity to the promoter region of target genes [41]. Factor Sp1 has also been studied in congenital heart defects; Li et al. for instance, studied the *NKX2.5* gene in murine models and, using a transcriptional regulatory networks analysis, found a dysregulation of Sp1 and other transcriptional factors, providing evidence of the Sp1 participation in the pathogenesis of CHDs [42]. Moreover, a relationship has been reported between Sp1 and other genes of the T-box family including the *TBX20* gene. Gong et al. for instance, evaluated DNA methylation levels in the promoter region of this gene in patients with TOF. The study provided evidence of the bond with Sp1 at the sequence TBX20_M1, which describes the influence of Sp1 in the expression of *TBX20*. This evidence suggests the participation of Sp1 in congenital septal defects [40].

It is important to highlight the main strengths of our study: (a) we included non-syndromic individuals, (b) the experimental strategy used was pyrosequencing, which is a quantitative method, (c) we analysed the exposure of environmental factors in mothers of patients with CHDs in order to determinate the possible association of the environment with the methylation levels of *TBX5* gene (this aspect has not been evaluated in previous studies of patients with congenital septal defects), and (d) we performed an in silico analysis that suggests binding of transcriptional factors to the CpG sequence studied. Interestingly, these transcriptional factors are related to cardiac pathologies. Nevertheless, limitations of present study should also be recognized: (a) we were not able to assess the functional effect of promoter methylation of the *TBX5* gene due to the impossibility of obtaining heart biopsies without clinical compromise, (b) we analysed six CpG sites in the promoter region, additional CpG sites are needed to establish the true role of methylation of the *TBX5* gene, (c) it would be interesting to address other regulatory regions and other epigenetic mechanisms of the *TBX5* gene, (d) it is advisable to expand the sample size in order to validate the present findings.

## 5. Conclusions

In conclusion, our data suggest that a high DNA methylation of the *TBX5* gene could be associated with congenital septal defects; additionally, environmental factors during pregnancy also confer a risk of developing congenital septal defects. These findings represent a first step to better understanding the epigenetic factors implicated in CHDs. Nevertheless, the exact molecular and epigenetic mechanisms remain unknown. Additional research in this field is necessary to provide new markers for clinical diagnosis in early onset of these pathologies.

## Figures and Tables

**Figure 1 biology-11-00096-f001:**
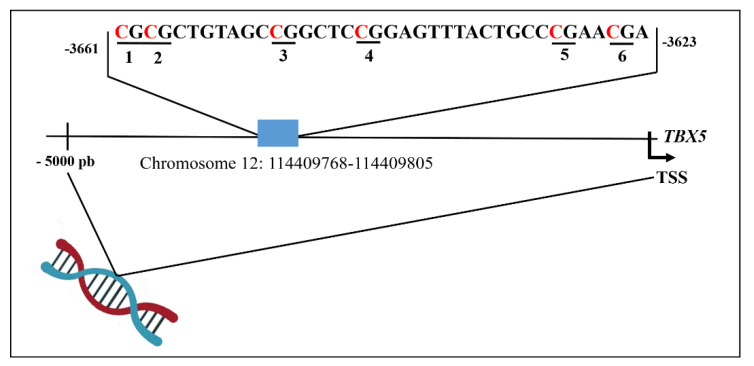
Schematic representation of distribution the CpG sites analysed in the *TBX5* gene promoter. Numbers 1-6 refer to CpG site within the sequence of *TBX5*. TSS, transcription start site.

**Figure 2 biology-11-00096-f002:**
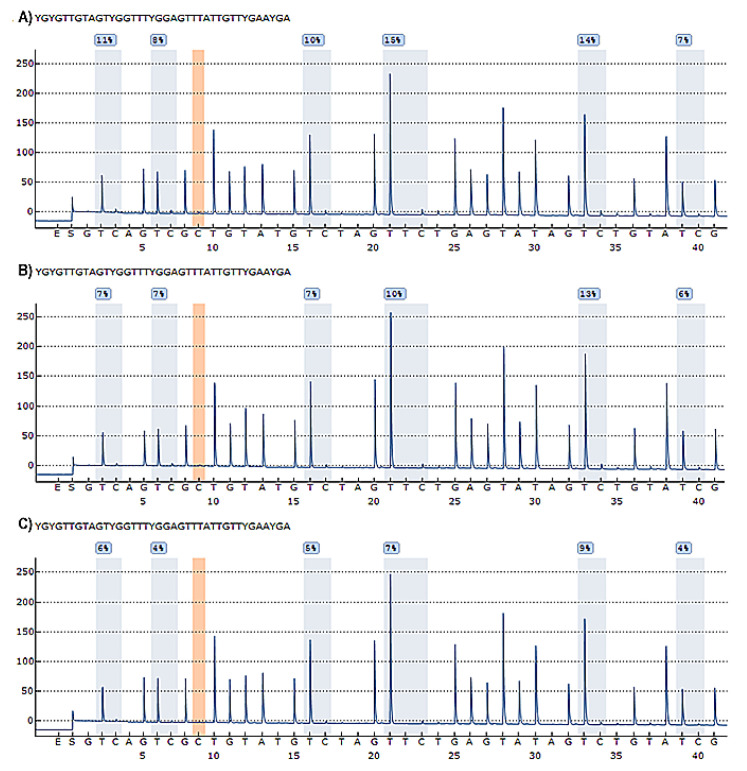
Pyrograms resulting from pyrosequencing process. (**A**) Pyrogram of patient with ASD; (**B**) Pyrogram of a patient with VSD; (**C**) Pyrogram of a control individual. In each pyrogram, the evaluated CpG sites are identified with the letter “Y”. The X-axis describes the order of dispensing during the pyrosequencing process, starting with the enzyme and the substrate, followed by the corresponding nucleotides; the Y-axis represents the relative light intensity corresponding to bioluminescence. The height of each peak in the CpG sites is proportional to the number of nucleotides incorporated in the analysed sequence. The blue shaded areas represent the CpG sites with their corresponding percentages of methylation; the presence of an orange shaded area indicates a control bisulfite dispensing, representing a suitable bisulfite conversion process.

**Figure 3 biology-11-00096-f003:**
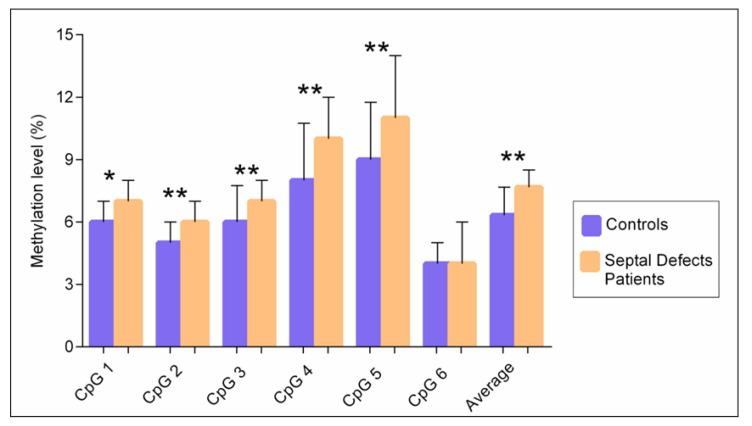
Median methylation levels for 6 CpG sites in the *TBX5* gene between patients with septal defects and the controls. * *p* < 0.05, ** *p* < 0.001; Mann–Whitney U test.

**Figure 4 biology-11-00096-f004:**
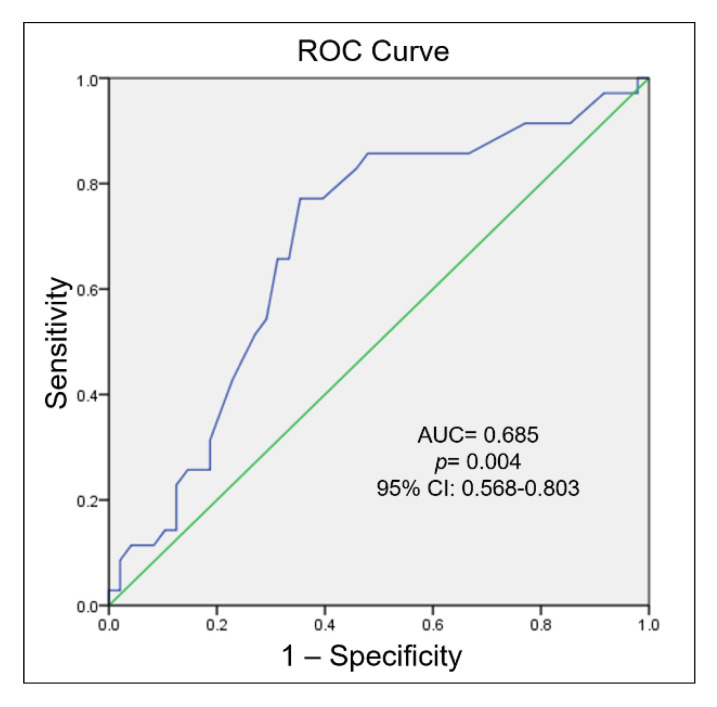
Receptor-operator characteristic (ROC) curve analysis associated to the presence of septal defects by *TBX5* gene methylation level.

**Figure 5 biology-11-00096-f005:**
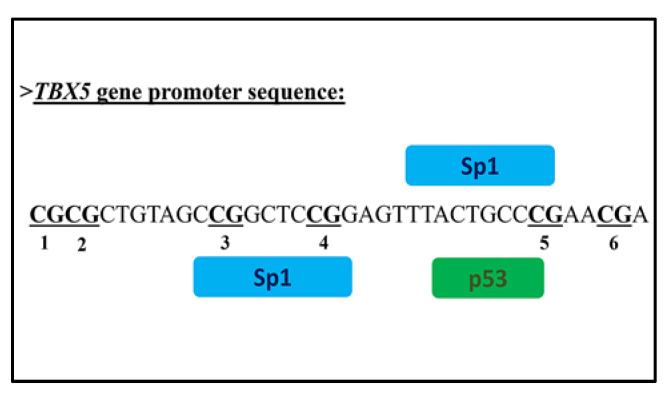
The potential binding sites of transcription factors identified with PROMO, TFBIND and AliBaba 2.1 software.

**Table 1 biology-11-00096-t001:** Demographic characteristics of the study population.

	Congenital Septal Defects Patients (n = 35)	Controls (n = 48)	*p* *
Age (in years) ^a^	8 (4–13)	3 (2–6)	0.005
Sex (%) ^b^			0.040
Females	17 (48.6)	34 (70.8)	
Males	18 (51.4)	14 (29.2)	

^a^ median [IQR], ^b^ n (%) for categorical variables. * Mann–Whitney U test for continues variables and χ^2^ test for categorical variables.

**Table 2 biology-11-00096-t002:** Association between *TBX5* methylation levels and risk of congenital heart defects.

*TBX5* Gene	Congenital Septal Defect Patients (n = 35)	Controls (n = 48)	OR (95% CI)	*p* *
Methylation Levels	n (%)	n (%)		
Highest quartile (>75%)	11 (31.4)	9 (18.8)	3.91 (1.02–14.8)	0.045
Medium quartile (25–75%)	19 (54.3)	23 (47.9)	2.64 (0.8–8.5)	0.105
Lowest quartile (<25%)	5 (14.3)	16 (33.3)	1.0 (Reference)	

OR, odds ratio, CI, confidence intervals. * Logistic regression.

**Table 3 biology-11-00096-t003:** Environmental factors analysed to determine the risk in the study population.

Environmental Risk Factors	OR (95% CI)	*p* *
Drug addiction	0.75 (0.26–2.15)	0.593
Diseases during pregnancy	1.77 (0.68–4.57)	0.236
Exposure to pollutants	1.14 (0.47–2.77)	0.765
Maternal infections	2.90 (1.01–8.33)	0.048
Medication consumption	2.03 (0.84–4.93)	0.115
Consumption of vitamins	0.22 (0.02–2.28)	0.208

OR, odds ratio, CI, confidence intervals. * Univariate logistic regression.

## Data Availability

All data are given in the main manuscript, or in the Appendix A.

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
