# Peer review of "DNA Methylation Levels of the TBX5 Gene Promoter Are Associated with Congenital Septal Defects in Mexican Paediatric Patients"

_biology, 2022, doi:10.3390/biology11010096_

Round 1

Reviewer 1 Report

The original research article entitled “DNA methylation levels of the TBX5 gene promoter are associated with congenital septal defects in Mexican pediatric patients” by Flores et al., evaluated the association between the methylation of TBX5 promoter and CHDs. Authors concluded that the high DNA methylation of the TBX5 gene is associated with congenital septal defects.

Here, authors proposed an interesting approach to identify the septal defects by methylation status. However, this reviewer have some concurs that might difficult the publication of this manuscript in the present form.

Please refer below comments.

Comments:

  1. Change the last sentence of the abstract as “Our data strongly suggest….”
  2. Authors should provide details, how they designed the methylation specific primers (MSPs).
  3. Authors should provide all relevant software’s used for primer design and data analysis.
  4. Elaborate the results section with details.
  5. Age differences were noticed among the controls and diseased groups. Authors should justify why they chose different age groups.  
  6. Discussion is a bit long and make it crisp.
  7. Page #8, line #309-310, reference missing for this statement in the discussion section.

Author Response

REVIEWER 1

We thank the reviewer for the time invested in reviewing our work. The observations made were very important to improve the manuscript.

Comments and suggestions for authors:

The original research article entitled “DNA methylation levels of the TBX5 gene promoter are associated with congenital septal defects in Mexican pediatric patients” by Flores et al., evaluated the association between the methylation of TBX5 promoter and CHDs. Authors concluded that the high DNA methylation of the TBX5 gene is associated with congenital septal defects.

Here, authors proposed an interesting approach to identify the septal defects by methylation status. However, this reviewer have some concurs that might difficult the publication of this manuscript in the present form.

Please refer below comments.

  1. Change the last sentence of the abstract as “Our data strongly suggest….”

Replay: In accordance with the reviewer, we changed the last sentence of the abstract:

We deleted the original sentence:

Our data highly suggest that a high DNA methylation of the TBX5 gene is associated with congenital septal defects.

We changed the sentence in the new corrected version of the manuscript:

“Our data suggest that a high DNA methylation of the TBX5 gene could be associated with congenital septal defects”.

  1. Authors should provide details, how they designed the methylation specific primers (MSPs).

Replay: In this study, we do not use MSPs to assess methylation status. Therefore, we did not have to design primers to analyze the CpG sites in the promoter region of the TBX5 gene. We used the pyrosequencing technique as experimental strategy, considering, that it is a quantitative method and we employed a design of PCR and sequencing set primers commercially available from the supplier (QIAGEN).

The information is included as described by the supplier:

Amplicon Length:

227 bp

Sequenced Strand:

Antisense

Biotin Modification on:

Reverse PCR primer

Chromosomal Location:

Chromosome 12, BP 114409768-114409805

Sequence to Analyze:

CGCGCTGTAGCCGGCTCCGGAGTTTACTGCCCGAACGA

Sequence After Bisulfate Treatment:

YGYGTTGTAGTYGGTTTYGGAGTTTATTGTTYGAAYGA

Number of CpG sites included:

6

Nucleotide dispensation order:

GTCGTCGTCTAGATAGTCGTTCGAGTAGTGTCGATCG

In order to clarify this point, the following information and sentence regarding the design assay was added in Material and Methods Section, 2.3 DNA methylation:

“Then, by the process of PCR and pyrosequencing the sequence after bisulfate treatment (YGYGTTGTAGTYGGTTTYGGAGTTTATTGTTYGAAYGA) was analyzed with the Hs_TBX5_04_PM 148 PyroMark CpG assay (GeneGlobe ID: PM000509609) commercially available from the supplier (QIAGEN, Hilden, Germany)” instead of “Then, for PCR and bisulfite pyrosequencing, pre-designed primers of Hs_TBX5_04_PM 148 PyroMark CpG assay were used (PM00050960).”

  1. Authors should provide all relevant software’s used for primer design and data analysis.

Replay: We do not use a specific software for primer design because we employed an assay design standardized and validated for the supplier QIAGEN commercially available. On the other hand, for the quantitative data analysis of methylation of each CpG site, we used the PyroMark Q24 Advanced Software V.3.0.0, which allowed us to analyze and visualize the methylation levels expressed as a percentage.

Now, this information was added in Material and Methods Section, in the point 2.3 DNA methylation.

“The pyrosequencing quality control was assessed for each sample using PyroMark Q24 Advanced Software V.3.0.0, which allowed us to analyze and visualize the methylation levels expressed as a percentage”

And the following information was deleted of the manuscript:

The pyrosequencing quality control was assessed for each sample using PyroMark Q24 Analysis Software (Qiagen v.2.0.2.5). The methylation quantification of each CpG site was expressed in percentages and was calculated using the PyroMarkTM Q24 Software. 

  1. Elaborate the results section with details.

Replay: We agree with the reviewer’s comment; with the aim of improving the interpretation of the methylation results obtained by the pyrosequencing technique, we have included some pyrograms, which contains the methylation levels of each CpG site expressed as percentage.

The following information and figure 2 was added in Results Section, 3.1. DNA methylation of the TBX5 gene promoter and its association with septal defects:

“Six CpG sites in the promoter region of the TBX5 gene were analyzed. We included an example of a pyrogram result, which contains the methylation quantification of each CpG site expressed as a percentage, in Figure 2”          

Figure 2.  Pyrograms resulting from pyrosequencing process. A) Pyrogram of patient with ASD; B) Pyrogram of patient with VSD; C) Pyrogram of a control individual. In each pyrogram the evaluated CpG sites are identified with the letter "Y". The X-axis describes the order of dispensing during the pyrosequencing process, starting with the enzyme and the subtracted, followed by the corresponding nucleotides; the Y-axis represents the relative light intensity corresponding to bioluminescence. The height of each peak in the CpG sites is proportional to the number of nucleotides incorporated in the analyzed sequence. The blue shaded areas represent the CpG sites with their corresponding percentages of methylation; the presence of an orange shaded area indicates a control bisulfite dispensing, representing a suitable bisulfite conversion process.

  1. Age differences were noticed among the controls and diseased groups. Authors should justify why they chose different age groups.  

Replay: The reviewer is right, with respect to the comment on the age differences between the study groups, this is mainly due to the fact that we only included pediatric patients from newborn to eighteen years with a single diagnosis of ASD, VSD and PDA without any syndromic forms of congenital heart disease. In this sense, these features are considered the main strength of our study.

In order to clarify this important issue, the following paragraph was added in Material and Methods Section, in the point 2.1 Individuals studied:

“The selection criteria of this study were individuals of both sexes including newborn to eighteen years with a single diagnosis non-syndromic of ASD, VSD and patent ductus ar-teriosus (PDA); the criteria exclusion were patients with another type of CHD, other types of heart malformations and other types of infectious heart diseases, additionally, all par-ticipants were assessed by expert clinical geneticists with the aim to exclude syndromic forms of CHD”

  1. Discussion is a bit long and make it crisp.

Replay: We have modified the discussion section to better contextualize the transcendence of our results.

To date, the epigenetic involvement of the TBX5 in CHDs is poorly understood. Thus, in this research, a quantitative DNA methylation analysis of the TBX5 gene promoter in non-syndromic patients with congenital septal defects was performed. Additionally, the mothers of individuals with septal defects were analyzed regarding their exposure to environmental risk factors during pregnancy; likewise, we made a bioinformatics analysis in the promoter region of TBX5 gene.

Our results showed significant differences in methylation levels when comparing patients with congenital septal defects to controls, which suggests that the TBX5 gene could be a marker for risk association with septal defects. This assertion is based on the results of the ROC analysis, suggesting that methylation levels may be an epigenetic marker to detect individuals with congenital septal defects.

As far as we know, this the first study with this approach that including congenital septal defect patients and TBX5 gene; however, various investigations have been carried out in patients with TOF, with different genes implicated in the cardiogenesis. TOF is a frequent type of CHDs with implications in four different malformations such as overriding aorta, pulmonic stenosis, ventricular septal defect (VSD) and right ventricular hypertrophy. Therefore, we consider pertinent TOF and the pathology included in our study share relationship such as septal defects, although authors include other genes relevants during the cardiogenesis [20,27-30].

For instance, Sheng et al. analyzed the methylation levels of 71 candidate genes for CHDs in 41 patients with TOF and 6 controls. The authors found significant differences in 26 genes and within these, the TBX5 gene showed increased methylation levels in patients with TOF when compared to controls [27]. Also, Zhang et al. reported that methylation levels of the RXRA gene promoter in 26 patients with TOF and 6 controls by means of quantitative methods. The authors found that the overall methylation levels in the promoter region, was significantly higher in patients with TOF than in controls [28]. In another research, Yuan et al. studied the methylation status of the VANGL2 gene in 15 patients with TOF and 5 controls. Their results showed that overall methylation levels of the VANGL2 were significantly higher in patients with TOF than in controls [20]. These results are in agreement with our findings, as we found that patients also presented high methylation levels than controls.

Evidence has shown that the fetus development could be influenced alterations in maternal DNA methylation; in this sense, the uterine microenvironment has the capacity to induce important changes in the global DNA methylation and can result in direct changes in gene expression in the developing fetus [31-35]. Therefore, we assessed environmental and lifestyle exposures factors during pregnancy in mothers of individuals with CHDs in order to understand this relationship. We observed DNA methylation level differences in individuals with CHDs when the mothers were exposed to harmful environmental factors during pregnancy. This evidence suggests at least in part, the participation of these epigenetic events on CHDs; nevertheless, the molecular mechanisms of these phenomena are still not fully understood.

The results of in silico analysis showed that modifications in the promoter region of the TBX5 gene produced binding sites for the transcription factors Sp1 (Specificity protein 1) and p53 (Tumor suppressor Trp53). Factor p53 is implicated in the transrepression and transactivation of multiple genes. The increased of p53 expression contributes mainly to cell cycle arrest, apoptosis, metabolism disorder and autophagy; it also plays a role in some cardiovascular diseases and CHDs, as it is required for embryonic heart development. During the embryonic and post-natal development, an inappropriate p53 activation can disrupt the development of the heart by initiating apoptosis and limiting cell proliferation [36,37]. Moreover, under physiological conditions, p53 participates in providing support to the cardiac structure, as well as in functions related to mechanisms of DNA repair, cell division, and cellular senescence [38,39].

Sp1 is a transcription factor implicated in several biological processes including cell growth, cell differentiation and chromatin remodeling [40]. This factor is able to increase or decrease the transcriptional activity, because of its affinity to the promoter region of target genes [41]. Factor Sp1 has also been studied in congenital heart defects; Li et al. for instance, studied the NKX2.5 gene in murine models and using a transcriptional regulatory networks analysis found a dysregulation of Sp1 and other transcriptional factors, providing evidence of the Sp1 participation in the pathogenesis of CHDs [42]. Moreover, it has been reported a relationship between Sp1 and other genes of the T-box family including the TBX20 gene. Gong et al. for instance, evaluated DNA methylation levels in the promoter region of this gene in patients with TOF. The study provided evidence of the bond with Sp1 at the sequence TBX20_M1, which describes the influence of Sp1 in the expression of TBX20. This evidence suggests the participation of Sp1 in congenital septal defects [40].

It is important to highlight the main strengths of our study: a) we included non-syndromic individuals, b) the experimental strategy used was pyrosequencing, which is a quantitative method, c) we analyzed the exposure of environmental factors in mothers of patients with CHDs in order to determinate the possible association of the environment with the methylation levels of TBX5 gene (this aspect has not been evaluated in previous studies of patients with congenital septal defects), and d) we performed an in silico analysis that suggests binding of transcriptional factors to the CpG sequence studied. Interestingly, these transcriptional factors are related to cardiac pathologies. Nevertheless, limitations of present study should also be recognized: a) we were not able to assess the functional effect of promoter methylation of TBX5 due to the impossibility of obtaining heart biopsies without clinical compromise, b) we analyzed six CpG sites in the promoter region, additional CpG sites are needed to establish the true role of methylation of TBX5, c) would be interesting to address other regulatory regions and other epigenetic mechanisms of the TBX5 gene, d) it is advisable to expand the sample size in order to validate the present findings.

  1. Page #8, line #309-310, reference missing for this statement in the discussion section.

Replay: Thanks to reviewer for the observation. In agreement with the reviewer in the previous question, we decided to eliminate the studies that were requested to be referenced, because they were not related to our main findings and furthermore, they made the discussion extensive and confusing.

Now, we deleted the references in the sentence:

 “Other studies have shown evidence of the participation of epigenetic and genetic mechanisms of the TBX5 gene in other types of CHDs [14, 17, 38, 20]” in the Discussion Section.

Reviewer 2 Report

The TBX5 gene regulates morphological changes during heart development. The authors evaluated TBX5 promoter methylation in 35 patients with congenital septal defects (CHD). They concluded that high DNA methylation of the six CpG islands of TBX5 promoter was associated with CHD. Although their statistical analysis was convincing and the results were extensively discussed, I suggest that they should further display the effects of TBX5 promoter methylation.

  1. The results of in silico analysis with PROMO and TFBIND, which predicts the possible functional effects for the TBX5 promoter region, should be exhibited in the “results” section.
  2. The TBX5 mRNA expression in patients should be detected by RT-qPCR to correlate the promoter methylation, transcription factors-binding, and gene expression.

Author Response

REVIEWER 2

We thank the reviewer for the time invested in reviewing our work. The observations made were very important to improve the manuscript.

Comments and suggestions for authors:

The TBX5 gene regulates morphological changes during heart development. The authors evaluated TBX5 promoter methylation in 35 patients with congenital septal defects (CHD). They concluded that high DNA methylation of the six CpG islands of TBX5 promoter was associated with CHD. Although their statistical analysis was convincing and the results were extensively discussed, I suggest that they should further display the effects of TBX5 promoter methylation.

  1. The results of in silico analysis with PROMO and TFBIND, which predicts the possible functional effects for the TBX5 promoter region, should be exhibited in the “results” section.

Replay: Following the Reviwer’s recommendation, we added the next information and figure from the in silico analysis in the Results section:

3.2. In silico analysis of transcription factor binding sites to TBX5 promoter region

The in silico analysis with PROMO, TFBIND and AliBaba 2.1 (http://gene-regulation.com/pub/programs/alibaba2/index.html) software for possible prediction of potential transcription factors binding sites to the TBX5 promoter region indicated the binding of Sp1 (Specificity protein 1) and p53 (Tumor suppressor Trp53) transcription factors. See Figure 5.

Figure 5. The potential binding sites of transcription factors identified with PROMO, TFBIND and AliBaba 2.1 software.

  1. The TBX5 mRNA expression in patients should be detected by RT-qPCR to correlate the promoter methylation, transcription factors-binding, and gene expression.

Replay: We totally agree with the Reviewer’s comment. In fact, the study, initially intended to correlate promoter methylation levels with gene expression in the heart. However, the size of the biopsy that could be obtained from certain patients without clinical compromise was not enough to perform the gene expression assays.

Currently, most patients undergo the correction of the septal defect using an Amplatzer device for transcatheter closure and thus avoid surgery, which has been associated with discomfort, morbidity, a thoracotomy scar and in some cases with complications. Thus, for these reasons we were not able to integrate the effects of TBX5 promoter methylation with gene expression. 

The aforementioned limitation has been recognized in the discussion section, page 10, lines 376-383 of the new corrected version of the manuscript as follows:

Nevertheless, limitations of present study should also be recognized: a) we were not able to assess the functional effect of promoter methylation of TBX5 due to the impossibility of obtaining heart biopsies without clinical compromise, b) we analyzed six CpG sites in the promoter region, additional CpG sites are needed to establish the true role of methylation of TBX5, c) would be interesting to address other regulatory regions and other epigenetic mechanisms of the TBX5 gene, d) it is advisable to expand the sample size in order to validate the present findings.

Reviewer 3 Report

Title: DNA methylation levels of the TBX5 gene promoter are associated with congenital septal defects in Mexican pediatric patients

Authors: Esbeidy García-Flores, José Manuel Rodríguez-Pérez, Verónica Marusa Borgonio-Cuadra, Gilberto Vargas-Alarcón, Juan Calderón-Colmenero, Juan Pablo Sandoval, José Antonio García-Montes, Víctor Manuel Espinoza-Gutiérrez, Juan Gerardo Reyes-García, Benny Giovanni Cazarín-Santos, Antonio Miranda-Duarte, Armando Gamboa-Domínguez and Nonanzit Pérez-Hernández.

General Comment:

Congenital heart defects (CHD) pathogenesis is multifactorial and involves both genetic and environmental elements. Epigenetic modifications appear to mediate between genes and the environment and therefore may play a decisive role in developing clinical outcomes. In their work, Esbeidy García-Flores et al. investigated the association of DNA methylation levels of the TBX5 gene promoter with the risk of developing congenital septal defects in the context of maternal exposure to environmental factors. While the research is innovative, the results can significantly contribute to a better understanding of CHD pathogenesis; some methodological concerns need to be resolved before the manuscript is accepted for publication.

Major revisions:

Materials and Methods

  • Does the decision to include the studied group participants with two different types of congenital septal defects (ASD and VSD) not hinder the interpretation of the results?

Were there any differences in the TBX5 promoter methylation pattern between participants with the two distinct congenital septal defects?

  • Shouldn't the control group consist of healthy individuals?

Results

  • Was TBX5 promoter methylation level an independent risk factor of congenital septal defects?
  • Please include the findings from the in silico analysis in the Results section.

Discussion

  • When discussing other studies on the contribution of epigenetic mechanisms to congenital septal defects, please make references to the study findings.
  • Please, mention some limitations of the study.

Minor revisions:

Abstract:

Please provide 95% CI for ORs.

Author Response

REVIEWER 3

We thank the reviewer for the time invested in reviewing our work. The observations made were very important to improve the manuscript.

Comments and suggestions for authors:

Congenital heart defects (CHD) pathogenesis is multifactorial and involves both genetic and environmental elements. Epigenetic modifications appear to mediate between genes and the environment and therefore may play a decisive role in developing clinical outcomes. In their work, Esbeidy García-Flores et al. investigated the association of DNA methylation levels of the TBX5 gene promoter with the risk of developing congenital septal defects in the context of maternal exposure to environmental factors. While the research is innovative, the results can significantly contribute to a better understanding of CHD pathogenesis; some methodological concerns need to be resolved before the manuscript is accepted for publication.

MAJOR REVISIONS:

Materials and Methods:

  1. Does the decision to include the studied group participants with two different types of congenital septal defects (ASD and VSD) not hinder the interpretation of the results?

Replay: The Reviewer’s comment is very pertinent. Even if both congenital septal defects seem to be different diseases, both conditions occur in related territories, the share the same characteristic of an incomplete septal closure and the same potential etiology. Therefore, both malformations lead to similar clinical complications, for example, pulmonary hypertension, cerebral infarction, and arrhythmias. Consequently, we were not concerned about data interpretations.

In order to clarify this important point, we added the following information in Materials and Methods Section, 2.1. Individuals studied:

For this study, we integrated these two types of CHDs (ASD and VSD) as group of patients, even if both congenital septal defects seem to be different diseases, both conditions occur in related territories, the share the same characteristic of an incomplete septal closure and the same potential etiology. Therefore, both malformations lead to similar clinical complications, for example, pulmonary hypertension, cerebral infarction, and arrhythmias [21,22].

  1. Were there any differences in the TBX5 promoter methylation pattern between participants with the two distinct congenital septal defects?

Replay: The Reviewer’s comment is very interesting; we analyzed whether there were differences between the TBX5 promoter methylation pattern of patients with congenital septal defects. This analysis showed that there were no statistically significant differences between both congenital septal defects in the average methylation levels and in the five CpG sites studied; except in the 6 CpG site. This unique difference probably was due to sample size of our study.

Bellow, we showed this analysis:

Table. Analysis of TBX5 promoter methylation between congenital septal defects (only for reviewer).

CpG Sites

 VSD

(11)

ASD

(24)

P value

1

7 (6-8)

7 (6.2-8)

0.442

2

6 (4-7)

6 (6-7)

0.495

3

7 (5-8)

7 (7-8.7)

0.410

4

9 (8-11)

10 (9-12)

0.233

5

10 (9-13)

11 (10-14)

0.181

6

0 (0-4)

5 (1.2-6)

0.028

Average

7 (6-8)

7.6- (7-8.7)

0.097

Data expressed as medians and interquartile range. P value were calculated by Mann-Whitney U test.

  1. Shouldn't the control group consist of healthy individuals?

Replay: We consider the Reviewer's comment very appropriate; most case-control studies evaluating the risk association use a control group consist of healthy individuals. In fact, the study, initially intended to correlate promoter methylation levels with gene expression in the heart. However, the size of the biopsy that could be obtained from certain patients without clinical compromise was not enough to perform the gene expression assays. In addition, the impossibility of obtained heart biopsies of the control group (healthy individuals), this considering the ethical implications and for this reason, a group of healthy individuals could not be used as a control group.

On the other hand, also have to consider, the control group is also known as a comparison group, in this sense and according to the literature (PMID: 20697313), is also valid compare with a group of individuals without the outcome of interest, who belong to the same population of origin with the same selection and exclusion criteria. Therefore, we consider to appropriate to use PDA as a control group.

 Results

  1. Was TBX5 promoter methylation level an independent risk factor of congenital septal defects?

Replay: Based on our results, until now it is difficult to conclude whether methylation levels are an independent risk factor of congenital septal defects. This considering that we studied the association between methylation levels and risk of congenital septal defects grouped in different quartiles using logistic regression analysis. However, this should be considered cautiously, due to the sample size and the lack of additional variables to remove the effect of confounding variables to avoid bias and thus establish whether this association is independent.

Considering the aforementioned, we have modified the last sentence of the abstract and conclusions sections:

We deleted the original sentence of the abstract section:

“Our data highly suggest that a high DNA methylation of the TBX5 gene is associated with congenital septal defects”.

We changed the sentence in the new corrected version of the abstract section:

“Our data suggest that a high DNA methylation of the TBX5 gene could be associated with con-genital septal defects”.

We deleted the original sentence of the first statement of conclusions section:

“In conclusion, a high DNA methylation of the TBX5 gene promoter is associated with congenital septal defects”.

We changed the sentence in the new corrected version of the first statement of conclusion section:

 “In conclusion, our data suggest that a high DNA methylation of the TBX5 gene could be associated with congenital septal defects”

  1. Please include the findings from the in silico analysis in the Results section.

3.2. In silico analysis of transcription factor binding sites to TBX5 promoter region.

The in silico analysis with PROMO, TFBIND and AliBaba 2.1 (http://gene-regulation.com/pub/programs/alibaba2/index.html) software for possible prediction of potential transcription factors binding sites to the TBX5 promoter region indicated the binding of Sp1 (Specificity protein 1) and p53 (Tumor suppressor Trp53) transcription factors. See Figure 5.

Figure 5. The potential binding sites of transcription factors identified with PROMO, TFBIND and AliBaba 2.1 software.

Discussion

  1. When discussing other studies on the contribution of epigenetic mechanisms to congenital septal defects, please make references to the study findings.

Replay: Thanks to reviewer for this interesting comment, we consider that the studies previously discussed were deleted, as the study design did not correspond to congenital septal defects. Therefore, the scope of the results was not relevant to our findings. Furthermore, in agreement with another reviewer, we decided to remove these references, due to the difficulty in discussing studies unrelated to our findings in congenital septal defects.

The following statements were deleted of discussion section:

“Other studies have shown evidence of the participation of epigenetic and genetic mechanisms of the TBX5 gene in other types CHDs, for instance, Ma et al. reported a hypermethylation of long-non coding RNAs (TBX5-AS1:2) which is implicated in a reduced TBX5 expression and participates in the regulation of cell proliferation in patients with TOF [14]. Besides, Lewandowski et al. showed in mouse embryos lacking histone deacetylase 3 (Hdac3), that this deacetylase interacts with the TBX5 gene to modulate acetylation, regulating the TBX5 activity during early cardiogenesis [17].

Wang et al. reported that a genetic variant of the TBX5 gene localized in the 3’UTR region (rs6489956) was associated with a high susceptibility of CHDs in two Chinese populations. Decreased TBX5 mRNA levels and the correlation with genotypes supported this finding. Additionally, functional analysis showed a binding affinity of the T allele with two miRNAs (miR-9 and miR-30a) [38]”

  1. Please, mention some limitations of the study.

Replay: We agree with the comment of reviewer, this point is very important. Now, we have added the following sentence in the discussion section:

“Nevertheless, limitations of present study should also be recognized: a) we were not able to assess the functional effect of promoter methylation of TBX5 due to the impossibility of obtaining heart biopsies without clinical compromise, b) we analyzed six CpG sites in the promoter region, additional CpG sites are needed to establish the true role of methylation of TBX5, c) would be interesting to address other regulatory regions and other epigenetic mechanisms of the TBX5 gene, d) it is advisable to expand the sample size in order to validate the present findings”

MINOR REVISIONS:

Abstract:

  1. Please provide 95% CI for ORs.

Replay: Thanks to reviewer for the observation.

We have added the information in the abstract section:

High methylation levels were associated with congenital septal defects (OR = 3.91; 95% CI = 1.02 – 14.8; P = 0.045). The analysis of Receiver Operating Characteristic (ROC) showed that the methylation levels of the TBX5 gene could be used as a risk marker for congenital septal defects (AUC = 0.68, 95% CI: 0.56 – 0.80; P = 0.004). Finally, the environmental factors analysis indicated that maternal infections increased the risk (OR = 2.90; 95% CI= 1.01-8.33; P = 0.048) of congenital septal defects. Our data suggest that a high DNA methylation of the TBX5 gene could be associated with congenital septal defects.

Round 2

Reviewer 1 Report

None

Reviewer 2 Report

The authors have answered my questions appropriately, and the manuscript has been improved. I accept its publication in Biology.

Reviewer 3 Report

I want to express my gratitude for the opportunity to re-review the paper entitled: "DNA methylation levels of the TBX5 gene promoter are associated with congenital septal defects in Mexican pediatric patients " by Esbeidy García-Flores et al. Since the authors addressed my concerns regarding the study design and manuscript structure, I find the work acceptable for publication in Biology.